# Spatiotemporal dynamics of tumor–CAR T-cell interaction following local administration in solid cancers

**Katherine Owens** [iD][1,2], **Aminur Rahman**[1,3], **Ivana Bozic** [iD][1,4*]

**1** Department of Applied Mathematics, University of Washington, Seattle, Washington, United States of America, **2** Vaccine and Infectious Disease Division, Fred Hutchinson Cancer Center, Seattle, Washington, United States of America, **3** Artificial Intelligence Institute in Dynamic Systems, University of Washington, Seattle, Washington, United States of America, **4** Public Health Sciences Division, Fred Hutchinson Cancer Center, Seattle, Washington, United States of America

* ibozic@uw.edu

**Data availability statement:** The murine data values used for model parameter estimation are

## Abstract

The success of chimeric antigen receptor (CAR) T-cell therapy in treating hematologic malignancies has generated widespread interest in translating this technology to solid cancers. However, issues like tumor infiltration, the immunosuppressive tumor microenvironment, and tumor heterogeneity limit its efficacy in the solid tumor setting. Recent experimental and clinical studies propose local administration directly into the tumor or at the tumor site to increase CAR T-cell infiltration and improve treatment outcomes. Characteristics of the types of solid tumors that may be the most receptive to this treatment approach remain unclear. In this work, we develop a simplified spatiotemporal model for CAR T-cell treatment of solid tumors, and use numerical simulations to compare the effect of introducing CAR T cells via intratumoral injection versus intracavitary administration in diverse cancer types. We demonstrate that the model can reproduce tumor and CAR T-cell data from small imaging studies of local administration of CAR T cells in mouse models. Our results suggest that locally administered CAR T cells will be most successful against slowly proliferating, highly diffusive tumors. In our simulations, assuming equal detectable tumor diameters at the time of treatment, low average tumor cell density is a better predictor of treatment success than total tumor burden or volume doubling time. These findings affirm the clinical observation that CAR T cells will not perform equally across different types of solid tumors, and suggest that measuring tumor density may be helpful when considering the feasibility of CAR T-cell therapy and planning dosages for a particular patient. We additionally find that local delivery of CAR T cells can result in deep tumor responses, provided that the initial CAR T-cell dose does not contain a significant fraction of exhausted cells.

available in the supporting data file. This file was created by digitizing data from images previously published by Skovgard et al., DOI: 10.1016/j.omto.2021.06.006 and Zhao et al., DOI: 10.1158/0008-5472.CAN-10-2880. MATLAB code used to produce the results included in this work is available on github at https://github.com/lacyk3/RadiallySymmetricCART.

**Funding:** This work was supported by the WiSTEM²D Scholar Award from Johnson & Johnson (to IB) and by the National Institute of Allergy and Infectious Diseases of the NIH award (T32AI118690 to KO). The funders had no role in study design, data collection and analysis, decision to publish, or preparation of the manuscript. The content is solely the responsibility of the authors and does not necessarily represent the official views of the NIH.

**Competing interests:** The authors have declared that no competing interests exist.

## Author summary

Injecting chimeric antigen receptor (CAR) T cells directly into solid tumors or their surrounding area may improve efficacy over systemic infusion. We develop a spatiotemporal mathematical model to simulate local CAR T-cell administration for treating a range of solid tumors. Our model reproduces kinetic tumor and CAR T-cell data from preclinical studies. Model analysis suggests it may be particularly promising to pursue CAR T-cell therapy for diffusive tumors, and it could be useful to measure tumor density when considering feasibility or planning dosages of CAR T-cell therapy for a particular patient. Simulations also suggest that using evaluation criteria that considers reduction in tumor density instead of diameter alone impacts the classification of patient response to intratumoral CAR T-cell injection.

## Introduction

CAR T-cell therapy involves genetically modifying a patient's T cells to better detect and mount an attack against cancer cells. This technology has demonstrated unprecedented success in combating previously untreatable hematologic malignancies, with response rates of 40-98%, and six drugs have been approved by the Federal Drug Administration since 2017 [1]. The success of CAR T-cell therapies against blood cancers has generated significant interest in adapting CAR T-cell technology to treat solid tumors [2,3]. In order to overcome the challenges presented by the complex biology of solid tumors, advances on three key fronts are being investigated: the specificity of CAR T cells [4], meaning their ability to recognize all tumor cells without targeting healthy cells, the long-term persistence of CAR T cells in the body [5,6], and the ability of CAR T cells to reach the tumor site and infiltrate the tumor [7,8]. A growing body of work has also focused on local delivery of CAR T cells, which can allow treatment to bypass physical barriers, shorten the distance traveled by T cells, and reduce on-target/off-tumor effects [9–12]. This could simultaneously increase the number of CAR T cells within the tumor and decrease the risk of immune-related toxicity. Expanding evidence from pre-clinical and phase I clinical trials also suggests that local delivery of CAR T cells is safe and feasible [13–20].

The efficacy of intracranial or intratumoral delivery of CAR T cells has been demonstrated in several mouse models of brain tumors [9–11,13], and in case studies in human patients [21–23]. Further studies in mice have tested regionally delivered CAR T cells against breast cancer and liver cancer [12,24], and in some cases demonstrated that regional delivery reduces the dose required for effective therapy compared with systemic administration [25,26]. A number of ongoing clinical trials utilize regional delivery of CAR T cells, with the majority targeting tumors of the central nervous system or liver [16–18,20,27]. Additional locoregional delivery trials target malignant pleural mesothelioma [15,28,29], head and neck cancers [30], and breast cancer [31].

Mathematical models of CAR T-cell treatment have enabled characterization of CAR T-cell population dynamics [32,33] and CAR T-cell proliferation and/or killing mechanisms [34–39], and have been used to predict patient outcomes [40–43] and suggest optimal treatment plans [42,44–49]. Tserunyan et al. and Nukula et al. provided comprehensive reviews on the subject [50,51]. Most existing models yield helpful insight under the assumption that cell populations are well-mixed throughout the course of treatment. However, this assumption is unlikely to hold during CAR T-cell treatment of solid tumors, where failure of CAR T cells to

infiltrate the tumor can be a mode of tumor escape [7]. Spatial variability is particularly relevant in the case of regional delivery of CAR T cells, where they may be injected directly into the center of the tumor or into a cavity containing the tumor. To account for spatial variation in cell concentration and provide insight into how the geometry of the delivery of CAR T cells impacts treatment efficacy, we employ a 3-D reaction-diffusion modeling framework for local administration of CAR T cells to treat solid tumors.

Reaction-diffusion equations have been used to model tumor growth for nearly three decades. Murray et al. pioneered this approach [52], writing out in words that

$$
\begin{array}{ccc}
\text{rate of change} & & \text{diffusion of cells (motility)} \\
\text{of tumor cell population} & = & + \\
& & \text{net cell proliferation}
\end{array}
$$

This formulation has inspired a rich body of work, particularly successful in modeling highly diffusive cancers like glioma, and their treatments including resection and chemotherapy [53–56]. Later models incorporating additional biological details like nutrient limitation [57–60] and non-uniform growth and migration caused by intrinsic and extrinsic factors [61–65], have built on this foundation to generate more realistic tumor growth patterns. A similar modeling paradigm was proposed by Matzavinos et al. [66] to study the immune response to a dormant tumor and Li et al. [67] to study the infiltration of T cells into breast cancer tumor cell clusters. Reaction-diffusion equations have further been used to model nanoparticle drug delivery and response [68–70], and to model intratumoral injection of ethanol [71,72]. However, the post-injection kinetics of adoptive cellular therapies like CAR T cells are fundamentally different than those of purely diffusive, small-molecule based drugs. As a "living drug," infused cells will ideally proliferate and persist in the body in a way that traditional pharmacokinetic modeling frameworks do not capture.

In this article, we develop a 3-dimensional spatio-temporal model for CAR T-cell therapies applied to dynamic solid tumors. We compare model simulations with experimental data from CAR T-cell imaging studies in mice to verify that model behavior is reasonable. We use this framework to compare two modes of locoregional delivery, intracavitary injection versus intratumoral injection. In particular, we discuss how tumor characteristics like growth rate and invasiveness impact the effectiveness of CAR T-cell treatment.

## Mathematical model

We developed a partial differential equations (PDE) model with thresholded diffusion, non-linear growth, and non-linear coupling to describe the spatial coevolution of tumor and CAR T cells in time. We use a density-dependent threshold for diffusion to reflect two modes of tumor growth in a unified model: a stationary phase, in which the tumor grows more dense but does not expand, and a diffusive phase, in which tumor cells both proliferate and spread. These two phases are rooted in observed phenotypic differences in the motility and proliferative capacity of cancer cells [73]. The "go-or-grow" hypothesis postulates that external factors such as low oxygen levels, chemoattractants and the rigidity of the extracellular matrix may cause cancer cells to switch between a less motile, highly proliferative state and a more invasive state [74]. In this simple initial model, we do not explicitly model these external factors, but assume that the transition to a more invasive state occurs when the tumor density at each spatial location surpasses a critical value. Additionally, we assume that CAR T cells can kill tumor cells upon interaction, proliferate when tumor cell lysis is occurring, become exhausted

through interaction with tumor cells, and senesce over time. Their movement is described by simple diffusion [75].

## Model equations

The density of tumor cells and CAR T cells at location $x$ and time $t$ are denoted by $u(x,t)$ and $v(x,t)$ respectively. The rate of change in the density of the tumor is the sum of density-dependent diffusion, and a forcing term $F_1$ that accounts for tumor growth in the absence of treatment and for the effect of CAR T cells on the tumor. We write the tumor concentration as

$$\frac{\partial u}{\partial t} = \nabla \cdot (D_T(u)\nabla u) + F_1(u,v); \qquad D_T = \begin{cases} 0 & u(\mathbf{x},t) \leq u^*, \\ D_T^* & u(\mathbf{x},t) > u^*. \end{cases} \qquad (1)$$

When the tumor density is below the tumor diffusion threshold, $u^*$, tumor cells do not diffuse. Once the tumor density at a spatial location exceeds the critical density, $u^*$, tumor cell density diffuses at a constant rate, $D_T^*$. A Dirichlet boundary condition of u = 0 at infinity is enforced, allowing unrestricted tumor expansion. Further details on boundary conditions of the PDE are included in the supplementary material, S1 Text page 3.

We constructed the forcing function, $F_1$, by adapting the ODE model studied by Owens and Bozic [45]. Let

$$F_1(u,v) = au(\mathbf{x},t)(1 - bu(\mathbf{x},t)) - D(\mathbf{x},t) \qquad (2)$$

where

$$D(\mathbf{x},t) = d\frac{(v(\mathbf{x},t)/u(\mathbf{x},t))^l}{s + (v(\mathbf{x},t)/u(\mathbf{x},t))^l}u(\mathbf{x},t).$$

Thus, in our model tumor growth is logistic, with proliferation rate $a$ and a local carrying capacity of $1/b$ for the density at each spatial point in the domain. Previous reaction-diffusion models of tumor growth have considered logistic growth, as well as exponential and gompertzian growth [53]. Tumor cells are killed by CAR T cells at a rate determined by the ratio of CAR T-cell to tumor density in a given location. Tumor cells can be killed at a maximum rate of $d$, with the half-maximal killing rate attained at the CAR T-cell to tumor ratio, $s$. The exponent, $l$, dictates the slope of the saturation curve, with larger values leading to increasingly "switch-like" behavior. This phenomenological functional form was proposed by de Pillis et al. in a model of adoptive cellular therapy, and found to best fit cytotoxicity assay data compared with mass action or traditional Michaelis-Menten kinetics [76].

The evolution of the CAR T cells is a diffusive process with the addition of a forcing function, $F_2$, that describes the interactions between tumor cells and CAR T cells [45]:

$$\frac{\partial v}{\partial t} = \nabla \cdot (D_C \nabla u) + F_2(u,v). \qquad (3)$$

CAR T cells diffuse at a constant rate $D_C$. A Neumann boundary condition is used to allow for proliferation at the edge of the domain without precipitous leakage, the details of which are reported in the supplementary material, S1 Text page 3.

The forcing function, $F_2$, was adapted from the ODE model studied by Owens and Bozic [45]. Let

$$F_2(u,v) = j\frac{D^2(\mathbf{x},t)}{k + D^2(\mathbf{x},t)}v(\mathbf{x},t) - mv(\mathbf{x},t) - qu(\mathbf{x},t)v(\mathbf{x},t), \qquad (4)$$

where, again,

$$D(\mathbf{x},t) = d \frac{(v(\mathbf{x},t)/u(\mathbf{x},t))^l}{s + (v(\mathbf{x},t)/u(\mathbf{x},t))^l} u(\mathbf{x},t). \tag{5}$$

The CAR T-cell proliferation term was also inspired by the model of tumor-immune interactions studied by de Pillis et al. [76]. With this functional form, CAR T-cell proliferation depends on cell lysis, $D(x,t)$. The maximum rate of cell proliferation is $j$, with half-maximal proliferation occurring when $D^2 = k$. Using a saturating term for the CAR T-cell proliferation rate imposes an upper limit, which is reasonable due to the biological constraints on cell division. This term differs from the recruitment term used by Owens and Bozic [45] because we do not include endogenous effector cells in this model. CAR T-cell density also decreases at rate $m$ due to cell death and at rate $q$ due to exhaustion upon repeated interaction with tumor cells.

In this work, we consider the evolution of a radially symmetric, spherical tumor growing on an infinite domain. This approach was also taken by Burgess et al. [54] in their three-dimensional extension of the work of Murray et al. modelling glioma [52]. Assuming spherical symmetry greatly reduces the complexity of the model, while still allowing us to test two medically relevant methods of local CAR T-cell administration: intracavitary injection (uniform concentration along the boundary of the tumor) and intratumoral injection (high concentration in the center of the tumor). This geometry could also be used to consider treatment plans involving multiple doses, combinations of intratumoral and intracavitary administration, and CAR T-cell injection following tumor resection. Illustrations of the initial CAR T-cell profile for the two administration methods that we will compare are given in Fig 1A and 1B. Applying this geometric consideration to equations (1) and (3) and keeping the boundary conditions as is, yields

$$\frac{\partial u}{\partial t} = \frac{1}{r^2} \frac{\partial}{\partial r}\left(D_T(u)r^2 \frac{\partial u}{\partial r}\right) + F_1(u,v); \qquad D_T = \begin{cases} 0 & u(r,t) \le u^*, \\ D_T^* & u(r,t) > u^*; \end{cases} \tag{6a}$$

and

$$\frac{\partial v}{\partial t} = \frac{1}{r^2} \frac{\partial}{\partial r}\left(D_C r^2 \frac{\partial v}{\partial r}\right) + F_2(u,v). \tag{6b}$$

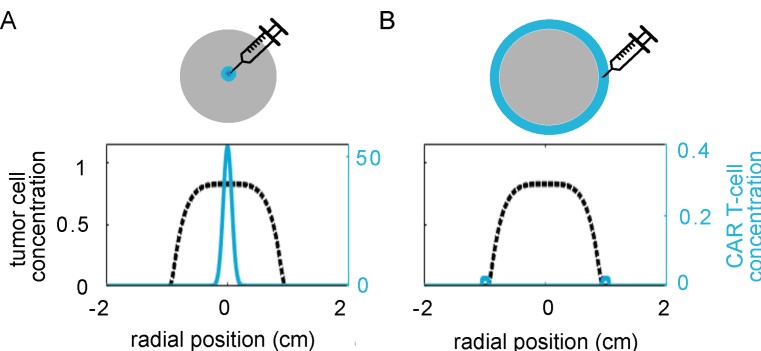

**Fig 1. Modes of local administration.** We model two possible methods of local administration: (**A**) intratumoral injection, initiated with a high concentration of CAR T cells at the center of the tumor mass and (**B**) intracavitary injection, initiated with a layer of CAR T cells on the surface of the tumor mass.

Our model incorporates the conversion of effector CAR T cells to an exhausted state through a mass-action exhaustion term in the CAR T-cell forcing function, $-quv$. In our reaction-diffusion model, CAR T cells are assumed to have an effector phenotype, hence all CAR T cells contribute to antitumor activity. However, experimental measurements of CAR T-cell kinetics over time do not discriminate between phenotypes. In order to compare simulation results with data, we add an ODE to the model to track the exhausted CAR T cells as well. Assuming that the exhausted CAR T cells have the same death rate as the effector CAR T cells, $m$, and an initial number of cells $v_E(0)$, the number of exhausted CAR T cells at time $t$ is given by

$$V_E(t) = V_E(0)\exp(-mt) + \int_0^t \exp(-m\tau)\int_0^{r_{max}} 4\pi q v(r,\tau)u(r,\tau)r\,dr\,d\tau. \qquad (7)$$

The total number of CAR T cells at time $t$ can then be calculated by summing the exhausted and the effector cells.

## Parameter estimation

In order to carry out informative numerical analysis and simulations of the model, we estimated several parameter values from experimental data and identified others from previous modeling work. We consider diffusion constants for tumor density in a range of $10^{-5}$–$10^{-4}$ cm$^2$/day. The upper end of this range is used to model diffusive tumors, and is calculated from measurements of the velocity of individual glioma cells *ex vivo* [77]. The diffusion constants for more compact tumors are an order of magnitude smaller, aligning with the range reported by Weis et al., who calibrated their reaction diffusion equation to breast cancer imaging data [78]. As the local carrying capacity for tumor cells, we use $2.39 \times 10^8$ cells/cm$^3$, based on assuming an average cell radius of 10 $\mu$m, as in previous reaction diffusion models of tumor growth [79,80]. The density above which diffusion of tumor cells occurs ranges from 1–50% of the local carrying capacity, allowing for a wide range of tumor density profiles. We use $10^8$ cells/cm$^3$ as the lower limit of detection when determining tumor radius for generating initial tumor conditions and evaluating the efficacy of treatment. This detection threshold is around 40% of the carrying capacity, which falls between the values used by Swanson et al. to model a highly sensitive imaging modality (16% of carrying capacity) and a less sensitive modality (80% of carrying capacity) [56]. The tumor proliferation rates studied here range from 0.025–0.25 day$^{-1}$, on the same order as previous models of tumor growth [45,54,55,78]. Particular values were selected to achieve tumor volume doubling times relevant to malignant gliomas (15–21 days) [81], breast cancer (46–825 days) [82], and liver cancer (85–150 days) [83] with a focus on the more aggressive end of each range because slow growing tumors are unlikely candidates for CAR T-cell therapy. We computed the diffusion constant for CAR T cells, $D_C = 1.38$ cm$^2$/day based on intratumoral CAR T-cell velocities reported from murine imaging studies by Mullazzani et al. [75]. The parameters dictating CAR T-cell proliferation, exhaustion, clearance, and killing of tumor cells came from the Patient 4 parameter set reported in Owens and Bozic [45]. To reflect the reduced efficacy of CAR T cells against solid tumors compared with blood cancers, we scaled these parameters, by 20%. The same approach was taken by Kara et al. [39] when adapting CAR T-cell models for blood cancers to solid tumors. We performed a local sensitivity analysis to study the effect of varying the CAR T-cell parameter values, which is discussed in the Results section. See Tables A and B in S1 Text for a complete list of parameter sets used in model simulations. The nondimensional version of the model is derived in the supplementary material, S1 Text page 2.

## Results

### Tumor growth in the absence of treatment

Our model provides a flexible framework to simulate a range of solid tumor behaviors. Here we consider four tumor types defined by a unique combination of a proliferation rate, $a$, diffusion coefficient, $D_T^*$, and diffusion threshold, $u^*$ (Table 1). For each tumor type, we generated a tumor density profile to represent the tumor burden at the time of CAR T-cell treatment. We initiated the simulation with a spherical tumor of radius 1 mm and uniform density at the carrying capacity, $1/b$, and then ran it forward in time until the detectable tumor radius reached 1 cm (Fig 2A). From these initial conditions, we simulated the progression of the tumor over the course of 200 days in the absence of treatment (Fig 2B). Tumor type I has a high proliferation rate and lower diffusivity resulting in a tumor with a volume doubling time (VDT) of 104 days and an almost uniform, highly dense profile. This type of growth pattern represents compact solid tumors, which could describe some liver or breast cancers [82,83]. Tumor type II has a moderate proliferation rate and moderate diffusivity, which results in a shorter VDT of 63 days and a slightly less compact cell density profile. This class could represent more aggressive forms of breast cancer [82]. Tumor type III has a low proliferation rate but high diffusivity, resulting in a VDT of 33 days. This balance of proliferation and diffusion generates an invasive but low density growth front that is able to advance undetected, behavior characteristic of a subset of gliomas [81]. Finally, tumor type IV is the most aggressive with both high proliferation and a high diffusivity. More advanced, aggressive gliomas may follow this growth pattern, which has a VDT of only 17 days [81].

We evaluated the impact of varying the tumor proliferation rate, tumor diffusion constant, and tumor diffusion threshold on VDT and the tumor burden at detection. VDT is more sensitive to the tumor diffusion constant than to the tumor proliferation rate (top row, Fig A in S1 Text). For tumors with a low proliferation rate and moderate to high diffusivity the tumor diffusion threshold modulates the extent of the tumor that is undetectable (bottom row, Fig A Panel A in S1 Text), whereas for tumors with higher proliferation rates the tumor burden at detection remains relatively constant despite changing the diffusion threshold (bottom row, Fig A Panel B-C in S1 Text).

### Intratumoral injection can lead to tumor eradication

We simulated intratumoral CAR T-cell injection by starting with a tumor of diameter 2 cm, constructed using growth parameters from one of the four tumor types defined in the previous section, and a high concentration of CAR T cells in the center of the tumor (see Materials and methods for details). We observe that CAR T cells injected intratumorally rapidly

**Table 1. Parameter values used in numerical simulations of the 4 tumor types. The source used to determine a reasonable value for each parameter is indicated in brackets next to each value. A fraction of the tumor carrying capacity (50%, 10%, or 1%) was chosen as the tumor diffusion threshold for each tumor type to obtain characteristic, qualitative behavior.**

| symbol | description | unit | Type I | Type II | Type III | Type IV |
|---|---|---|---|---|---|---|
| $a$ | tumor proliferation rate | day$^{-1}$ | 0.25 [82,83] | 0.125 [82] | 0.025 [81] | 0.25 [81] |
| $D_T^*$ | tumor diffusion constant | cm$^2$/day | 1e-5 [78] | 2e-5 [78] | 1e-4 [55] | 1e-4 [55] |
| $u^*$ | tumor diffusion threshold | cells/cm$^3$ | 1.2e8 | 2.39e7 | 2.39e6 | 2.39e6 |

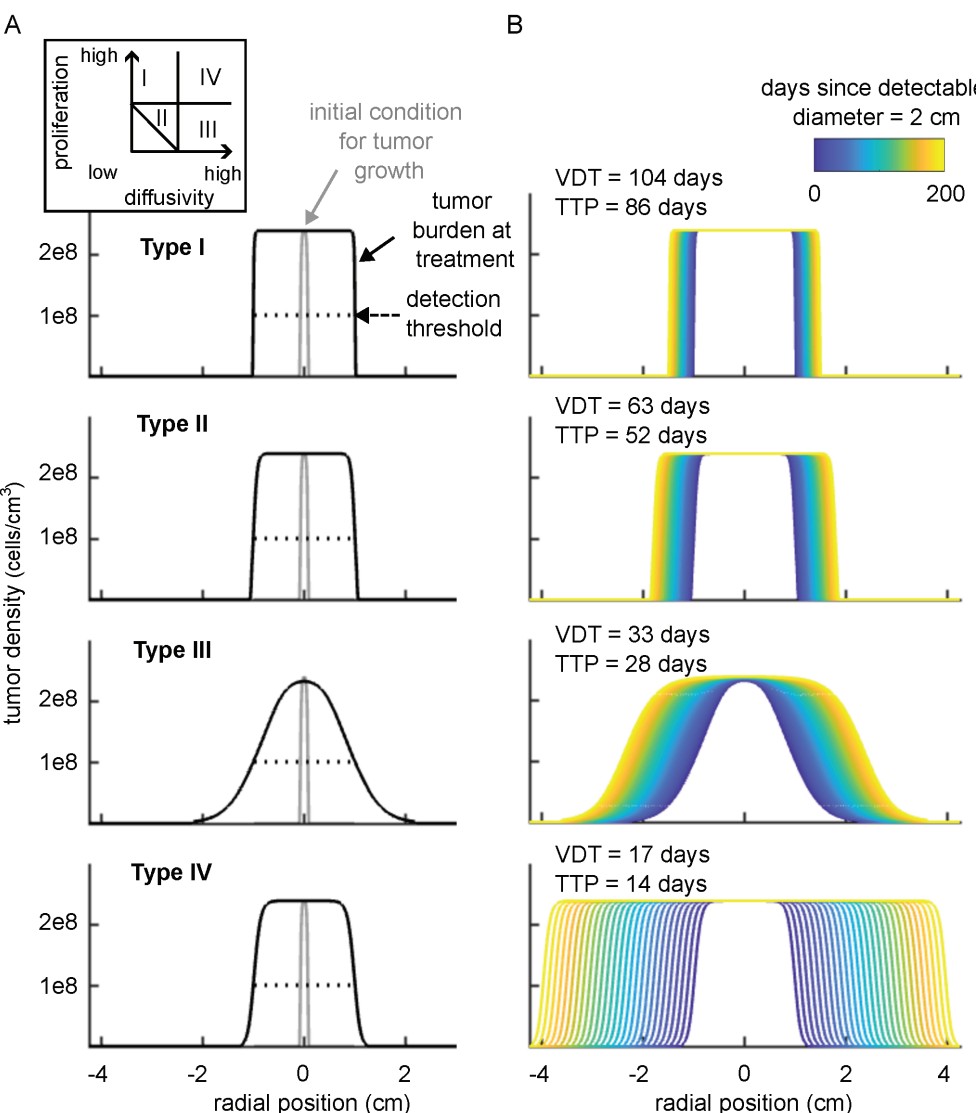

**Fig 2. Tumor growth in the absence of treatment for 4 tumor types.** (**A**) Density profile for each of these tumors when initiated from a 1 mm mass (gray line) and allowed to grow until the detectable diameter reaches 2 cm (black line) when using a detection threshold around $10^8$ cells/cm$^3$. If these four tumors remain untreated, their expansion would continue. (**B**) We illustrate the growth of the untreated tumors over 200 days. The volume doubling time (VDT) and time to progression (TTP) were calculated from the tumor with detectable diameter of 2 cm as a baseline. As diagrammed in the inset, Type I proliferates rapidly and diffuses slowly, Type II both proliferates and diffuses at a moderate rate, Type III proliferates slowly but diffuses rapidly, and Type IV both proliferates and diffuses rapidly.

kill cells at the center of the tumor and start to diffuse out towards the surface, both proliferating and killing tumor cells as they travel (Fig 3A). The outcome of treatment strongly depends on whether CAR T cells can proliferate and diffuse sufficiently to reach the surface of the tumor before becoming exhausted. In numerical simulations, we observe four potential treatment outcomes, defined by clinical criteria of response in solid tumors [84]. At sufficiently large doses of effector CAR T cells, intratumoral injection eliminates the tumor, leading to complete response (Fig 3A and 3B). For tumor types I-III a partial response is possible in which the CAR T cells reduce the detectable tumor radius significantly, but relapse occurs

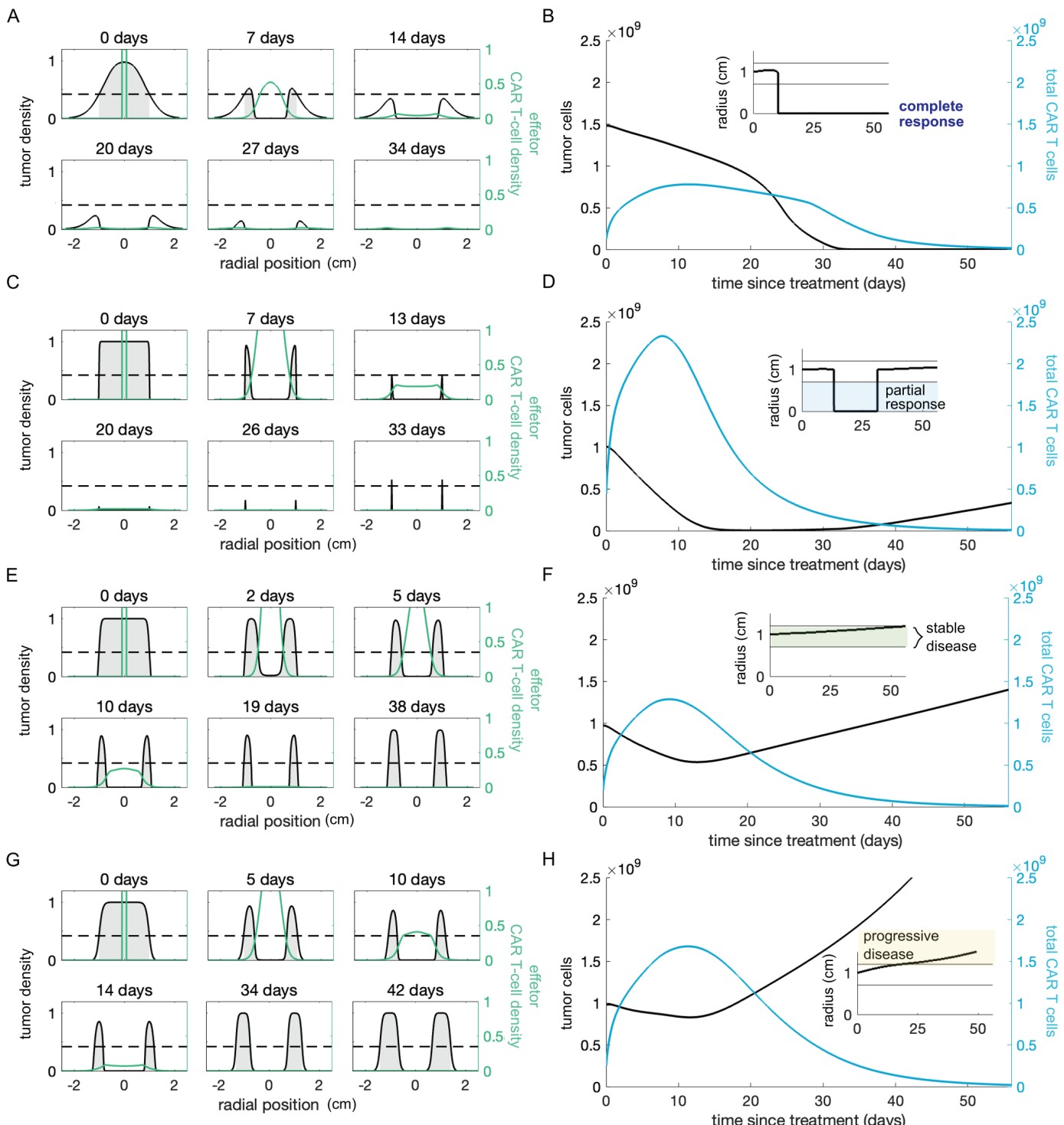

**Fig 3. Intratumoral treatment outcomes.** The first column shows snapshots of the cell density profiles following CAR T-cell injection. For each of these examples we assumed that 100% of the injected CAR T cells are effectors, and only the density of effector CAR T cells is depicted. The tumor detection threshold is indicated with a dotted line and the portion of the tumor that is detectable is shaded in gray. The second column shows the tumor and total CAR T-cell population (effector + exhausted) over time with the corresponding detectable tumor radius inset. (**A–B**) Intratumoral injection of $1 \times 10^9$ CAR T cells to treat tumor type III eradicates the tumor around day 30 post-injection.(**C–D**) Treating tumor type I with an intratumoral dose of $1 \times 10^9$ CAR T cells reduces the detectable tumor burden and total tumor cell count; however, the tumor resumes growing once CAR T cells dissipate. (**E–F**) Intratumoral injection of $2 \times 10^8$ CAR T cells to treat tumor type II initially reduces the tumor cell count, causing the detectable diameter to stay within the size classified as stable over 8 weeks following treatment. However, ultimately this tumor escapes. (**G–H**) Treating tumor type IV with $2 \times 10^8$ cells, kills tumor cells at the center of the initial mass, but the tumor continues to grow uncontrolled.

rapidly after effector CAR T cells dissipate allowing tumor cells at the edge of the initial tumor to proliferate (Fig 3C and 3D). Such rapid relapse following CAR T-cell treatment has been observed in the clinic for hematological malignancies. For example, 49% percent of relapses after CAR T-cell treatment in relapsed/refractory DLBCL occur within the first month [85]. With lower doses, CAR T cells may proliferate transiently but allow tumor escape. In these cases, CAR T cells reduce tumor density at the core, but the detectable tumor radius remains unaffected because the T cells do not reach the surface of the tumor in sufficient numbers to contain tumor growth. If the tumor is slow-growing, tumor progression may appear to be stable at an 8 week follow-up despite eventual escape (Fig 3E and 3F). With more aggressive tumors, disease progression following failed treatment occurs even faster (Fig 3G and 3H). For even lower doses of CAR T cells, the injected CAR T cells barely proliferate, instead dying off with little to no noticeable impact on the tumor. Failure of CAR T cells to successfully engraft and proliferate is also a documented failure mode, particularly in the hostile solid tumor micro-environment [86].

## Intracavitary delivery can delay tumor progression

Intracavitary administration of CAR T cells involves delivering CAR T cells into the region surrounding a tumor, but not directly into the tumor itself. For example, in one patient with recurrent glioblastoma, CAR T cells were infused intracranially directly through a catheter device [21]. Another example is delivering CAR T cells via hepatic artery infusion to treat tumors in the liver [27]. In contrast with intratumoral injection, in our model simulations CAR T cells delivered intracavitarily contain the tumor at the growth front as they migrate inwards across the tumor from the outer edge towards the center. To simulate intracavitary delivery of CAR T cells, we initiate the system with one of the four final tumor profiles from Fig 2A, and a thin layer of CAR T cells uniformly concentrated on the surface of the tumor. Using a uniform, symmetrical initial distribution of CAR T cells is a heavily simplified, numerically expedient way to reflect the evidence from imaging studies showing that CAR T cells rapidly disperse across solid tumors [75].

In our simulations using realistic clinical doses, CAR T cells delivered intracavitarily never eradicate tumors with an initial diameter of 2 cm or more. However, sufficiently large doses can reduce the detectable tumor radius of tumor types I and II, and consequently delay the time to tumor progression. An example of one such scenario is depicted in Fig 4A and 4B. Against tumor types II and III, moderate to large doses of CAR T cells delivered intracavitarily can hold the tumor to a steady size for an extended period of time. For tumor type III, large doses markedly change the density profile of the tumor, making it more compact (Fig 4C and 4D). At low doses against all tumors, or even moderate doses against the most aggressive tumors, e.g., type IV, intracavitary delivery of CAR T cells has a minimal impact on tumor progression (Fig 4E and 4F).

## Model can reproduce data from experiments in mice

After establishing the general behavior of our model, we verified that the model could be fit to data from early imaging studies in mice. We digitized data from two studies: one testing the efficacy of CAR T cells injected directly into solid tumors [87] and another comparing the efficacy of regional vs. systemic delivery of CAR T cells in treating solid tumors [88].

Zhao et al. established large flank mesothelioma tumors in mice and then treated them with intratumoral injections of CAR T cells [87]. In most of their experiments the animals received multiple CAR T-cell injections; however, the authors included data from one mouse

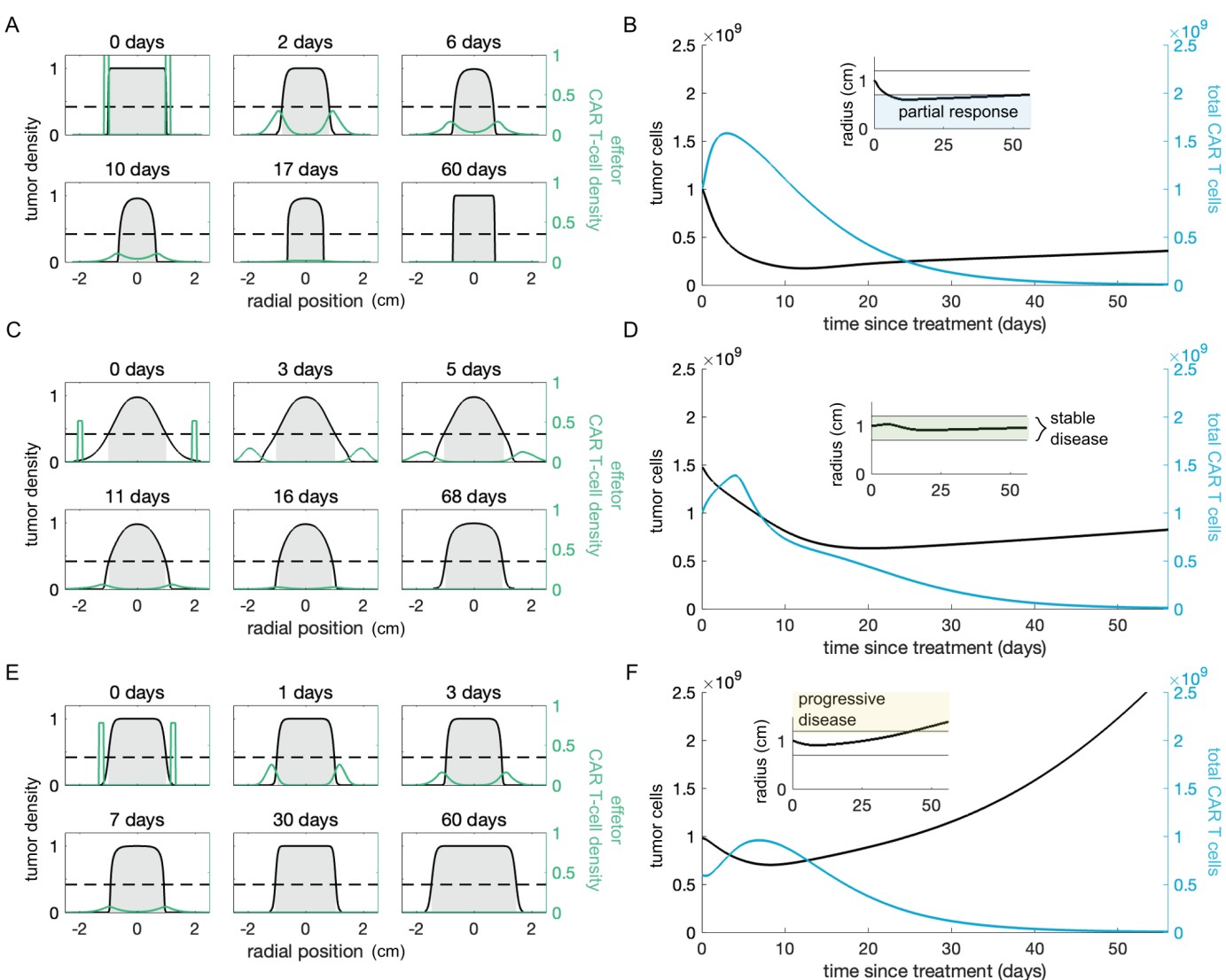

**Fig 4. Intracavitary treatment outcomes.** The first column shows snapshots of the cell density profiles over time. Only the density of effector CAR T cells is depicted. The detection threshold is indicated with a dotted line and the portion of the tumor that is detectable is shaded in gray. The second column shows the tumor and total CAR T-cell population (effector + exhausted) over time with the corresponding detectable tumor radius inset. For each of these examples we assumed that 100% of the injected CAR T cells are effectors. (**A–B**) Treating tumor type I with an intratumoral dose of $1 \times 10^9$ CAR T cells reduces the detectable tumor burden and total tumor cell count; however, the tumor resumes growing once CAR T cells dissipate. (**C–D**) Intratumoral injection of $1 \times 10^9$ CAR T cells to treat tumor type III holds the tumor cell count and detectable diameter steady over 8 weeks following treatment. (**E–F**) Treating tumor type IV with $2 \times 10^8$ cells, kills tumor cells at the edge of the initial mass, but the tumor continues to grow uncontrolled.

that cleared its tumor after receiving only one injection (Fig 5A). We fit our model to the corresponding longitudinal measurements of tumor size. With appropriate parameter values, our intratumoral injection model accurately simulates the change in tumor size over time, showing tumor eradication around day 15 (Fig 5B). Without treatment, our model predicts that the tumor would have tripled in size over baseline by day 18.

Skovgard et al. assessed the effect of antigen expression density on CAR T-cell kinetics and compared the efficacy of systemic vs. intracavitary delivery of CAR T cells [88]. The authors established antigen-positive orthotopic mesothelioma tumors in mice, and then transferred

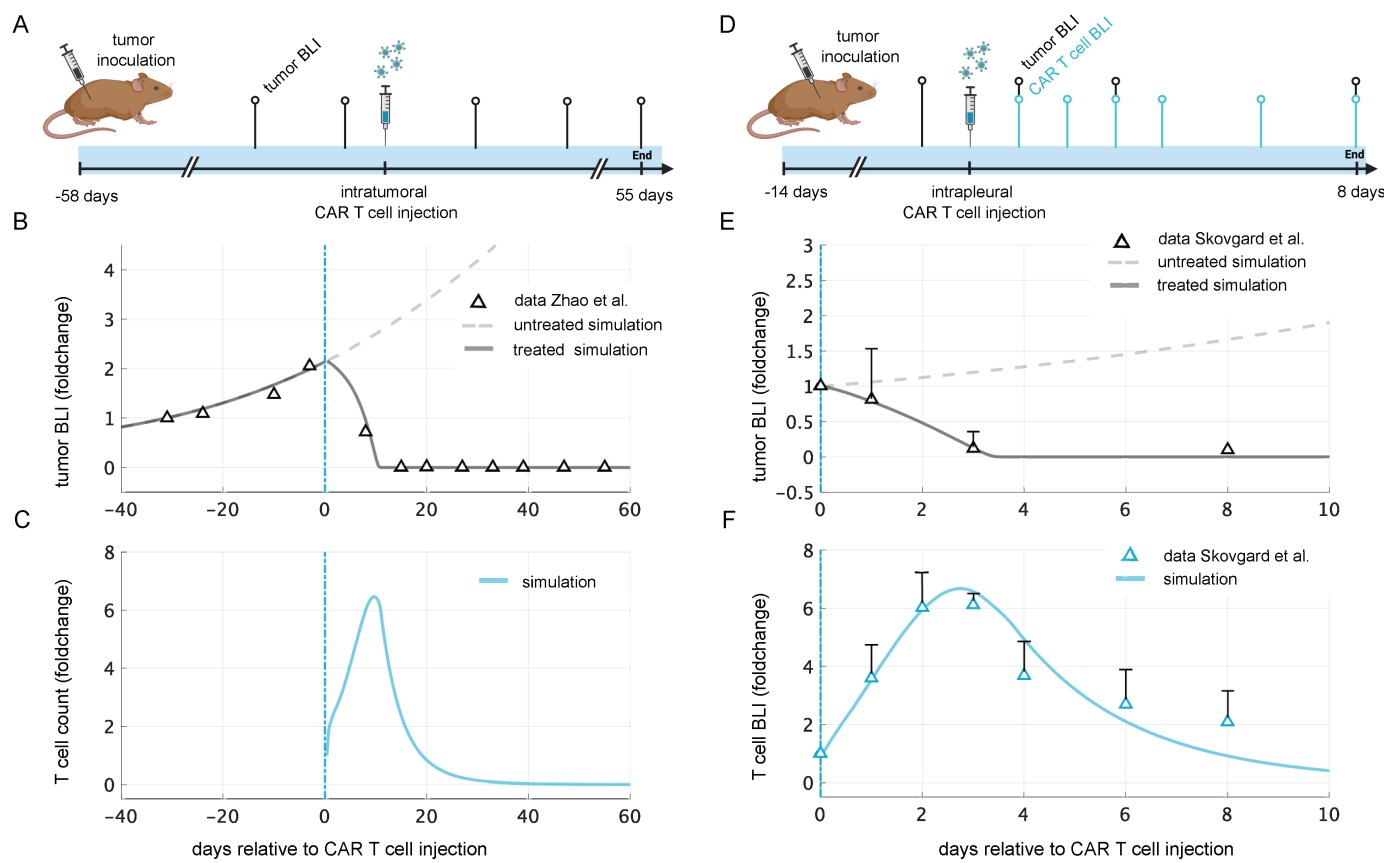

**Fig 5. Model fit to data from murine studies.** (**A**) Zhao et al. [87] established large flank mesothelioma tumors in mice, measured baseline tumor size using bioluminescent imaging (BLI), injected $1 \times 10^7$ CAR T cells intratumorally 58 days following tumor establishment, and continued to measure the tumor size using BLI for 55 days. (**B**) Model simulation of untreated tumor and treated tumor overlaid with treated tumor data from Zhao et al. [87] (**C**) The corresponding CAR T-cell count predicted by the model simulation. (**D**) Skovgard et al. [88] established small antigen-positive orthotopic mesothelioma tumors in mice, measured baseline tumor size using bioluminescent imaging (BLI), injected $1 \times 10^7$ CAR T cells intrapleurally 14 days following tumor establishment, and continued to measure the tumor size and CAR T-cell quantity for 8 days using BLI. (**E**) Model simulation of untreated tumor and treated tumor overlaid with treated tumor BLI quantification from Skovgard et al. [88] (**F**) The model simulation of CAR T-cell count is shown overlaid with CAR T-cell BLI quantification from Skovgard et al. [88]. Created in BioRender. Owens, K. (2025) https://BioRender.com/e999s6b

CAR T cells into the mice via direct pleural injection into the thoracic cavity (Fig 5D). They monitored CAR T-cell quantity in this group of 5 mice via bioluminescent imaging (BLI) for 8 days following injection. A separate group consisting of 7 mice treated in the same manner was used to measure tumor size via BLI over the same period. We fit our model to the mean value of these groups. Our intracavitary injection model shows tumor eradication by day 4, accompanied by a 6-fold increase in CAR T cells over their baseline measurement, in good agreement with the reported tumor and CAR T-cell levels (Fig 5E and 5F). Including the exhausted CAR T cells in the total projected CAR T-cell numbers achieves reasonable agreement with the data, though the model simulation still falls below the observed data points at 6 and 8 days post CAR T-cell injection (Fig B in S1 Text).

For an explanation of the parameter estimation procedure used to obtain these model fits, see the Materials and Methods section. These two examples provide proof-of-concept that the model can be fit to data from imaging studies. As additional data becomes available, a more

robust parameter estimation procedure could be developed to better quantify tumor-CAR T-cell dynamics.

### *Ex vivo* CAR T-cell exhaustion can result in treatment failure

We next used our model to evaluate the impact of *ex vivo* exhaustion on the efficacy of CAR T-cell therapy given different tumor characteristics and treatment delivery modes. We tested simulated CAR T cells administered either at the center of the tumor or the surface against the four tumor types summarized in Table 1 and illustrated in Fig 2 with CAR T-cell doses from $1 \times 10^7$ to $1 \times 10^9$ cells, spanning a range of reasonable dose levels [27]. Within that dose, we varied the fraction of the CAR T-cell product that is exhausted prior to administration from 0–100%, based on analysis by Long et al. [89]. We assumed a detectable diameter of 2 cm at the time of treatment, and simulations were run out to 8 weeks post treatment, at which point the patient response to treatment was classified according to the Response Evaluation Criteria in Solid Tumours (RECIST) [84]. The RECIST criteria is a standardized set of guidelines widely used to assess the efficacy of cancer treatments against solid tumors. Under these guidelines tumor response is classified as complete response if all measurable lesions disappear, a partial response if measurable lesions decrease in diameter by at least 30%, or progressive disease if measurable lesions increase in diameter by 20% or new lesions form. Otherwise the response is deemed stable disease.

Against all four tumors tested here, intratumoral injection of CAR T cells can eradicate the tumor at high enough doses (Fig 6, row 1), whereas the best possible outcome from intracavitary delivery is a partial response for tumor types I and II, or stable disease for tumor types III and IV (Fig 6, row 2). However, otherwise effective doses may fail if the percentage of CAR T cells that are exhausted ex vivo, prior to injection, is too high. For example, an intratumoral injection of $3 \times 10^8$ CAR T cells generates a complete response against tumor type IV if 90% of the cells are fully functional, but this same dose is predicted to result in progressive disease if more than 20% of the cells are exhausted (Fig 6, top right).

We also tested the effect of tumor size at the time of CAR T-cell administration on patient outcome by performing these same tests on tumors with detectable diameters of 1 cm and 3 cm at treatment. Our simulations show smaller tumors can be eradicated faster, and with lower doses than larger tumors of the same type (Fig C in S1 Text). When treating small tumors, intracavitary administration can induce a complete response for tumor types I–III (Fig C in S1 Text). When treating larger tumors, intratumoral injection no longer eradicates tumor type IV even at the highest doses tested (Fig C in S1 Text). Note that, unsurprisingly, the timing of evaluation impacts which dose levels are identified as inducing stable disease. Early evaluation can result in a classification of stable disease at lower doses for some scenarios (Fig D in S1 Text) and conversely later evaluation requires a higher dose to stabilize tumors, if it is even possible (Fig D in S1 Text).

Assuming a constant detectable tumor radius of 2 cm at the time of treatment, we examined the relationship between quantitative tumor characteristics and the minimum intratumoral CAR T-cell dose necessary for tumor eradication. The density of the detectable tumor and the ratio of tumor growth rate to tumor diffusion constant were both reasonably good predictors of the minimum successful CAR T-cell dose, with a coefficient of determination of $R^2 = 0.83$ and $R^2 = 0.84$ respectively and Akaike Information Criterion differing by 1 (Fig E Panel A and B in S1 Text). Previous work with a similar model for glioma growth identified that this parameter ratio dictates tumor density [55]. In contrast, for a fixed detectable

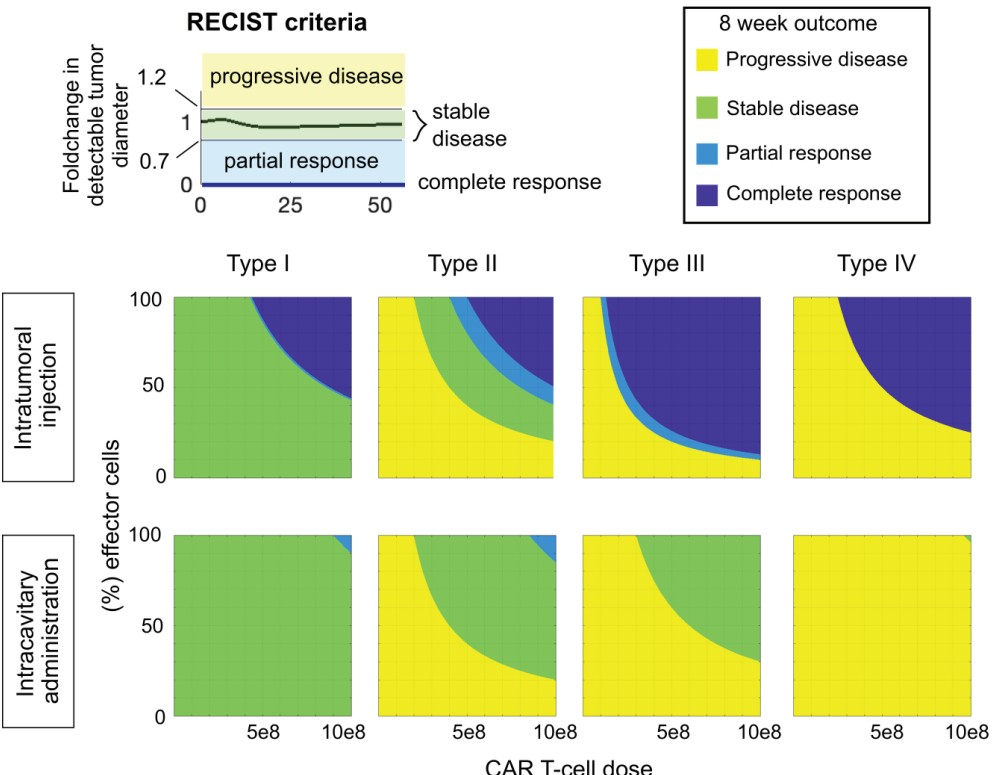

**Fig 6. Simulated treatment outcomes vary across different modes of delivery, tumor types, dose levels, and proportions of exhausted cells.** Each row corresponds to a mode of local delivery and each column corresponds to a different tumor type as defined in Table 1 and illustrated in Fig 2, with type I having the longest volume doubling time and type IV having the shortest volume doubling time. Each panel maps a range of CAR T-cell doses and percentage of non-exhausted cells within that dose to patient outcome, classified using the RECIST criteria [84] at 8 weeks post-treatment. In each simulation, CAR T-cell treatment occurred when the detectable tumor radius reached 2 cm.

tumor diameter at the time of treatment, the total tumor burden, including undetectable portions of the tumor, and the volume doubling time explain less than 40% of the variance in the minimum CAR T-cell dose needed for successful treatment (Fig E Panel C–D S1 Text).

## Choice of evaluation criteria impacts classification of partial response and stable disease outcomes

The RECIST criteria for classifying solid tumor response is based solely on the number of tumor lesions and their detectable diameter. Consequently, a treatment that drastically reduces density of the tumor interior but does not impact the extent of the tumor may be falsely classified as completely ineffective (Fig 7A). As an alternative, we evaluated treatment outcomes for the same scenarios depicted in Fig 6 using the Choi criteria [90], which assigns a label of partial response if either a density-based or a diameter-based threshold for tumor burden reduction is met (Fig 7B and 7C). The minimum dose level to achieve a partial response upon intratumoral injection is lower for all four tumor types, especially the less diffusive tumors, Type I and II (Fig 7D). For intracavitary administration, the alternate evaluation criteria does not make a significant difference because partial tumor burden reduction occurs at the growth front of the tumor and causes a reduction in detectable tumor diameter.

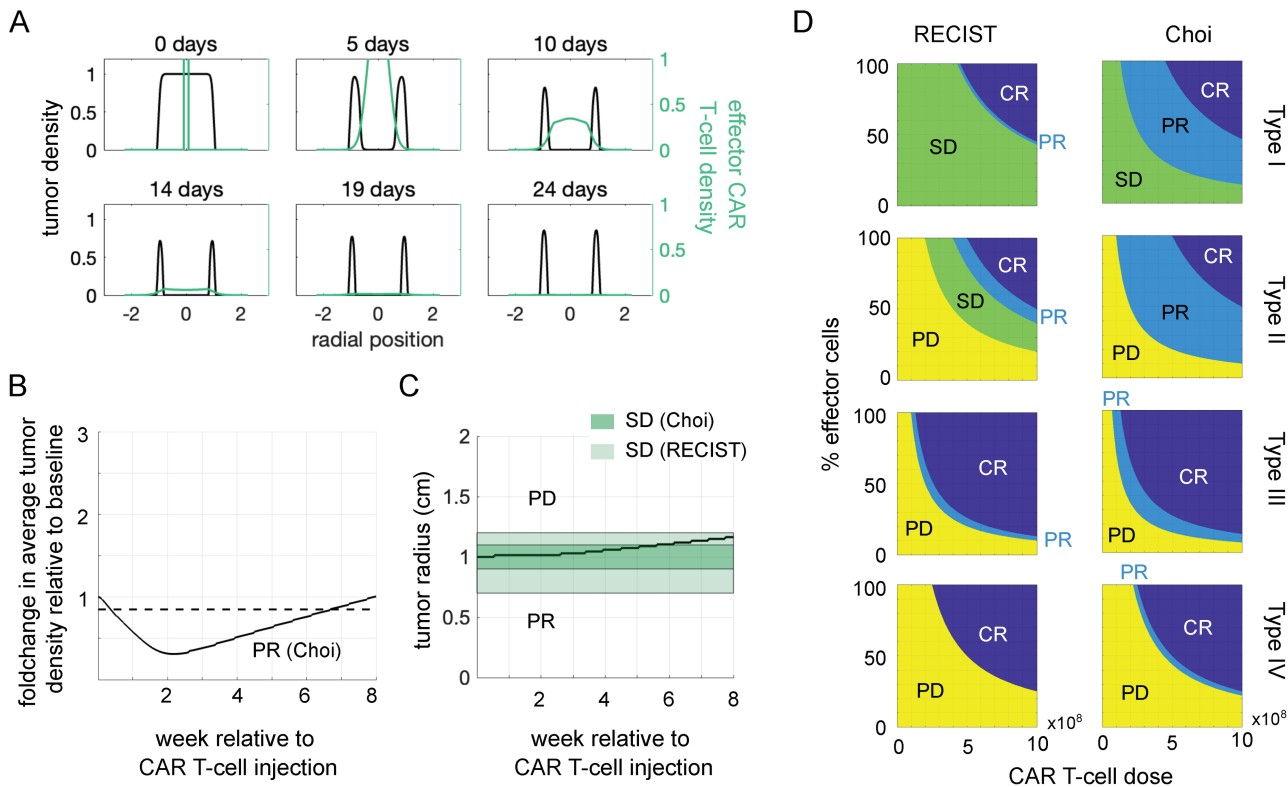

**Fig 7. Classification of treatment outcome following intratumoral CAR T-cell treatment evaluated using Choi versus RECIST criteria. (A)** Tumor and effector CAR T-cell densities over the first 24 days following intratumoral injection of $3 \times 10^8$ effector CAR T cells to treat a 2 cm diameter tumor of type II show wide variation across the spatial domain. (**B**) The fold-change in average tumor density over time reflects the effect of CAR T cells killing tumor cells at the tumor core. Because tumor burden drops below the horizontal dotted line at 0.85, the outcome is classified as partial response to therapy by the Choi criteria. (**C**) The detectable tumor burden for the same scenario increased monotonically. The size-based criteria for stable disease according to the Choi criteria requires staying in the region highlighted in dark green, which this scenario does not meet. The boundaries for stable disease according to the RECIST criteria are highlighted in light green, which this scenario does meet. (**D**) Outcome maps if four tumor types (row 1-4) were treated with intratumoral injection of a range CAR T-cell doses and percentage of non-exhausted cells within that dose, and evaluated at 8 weeks post-injection via the RECIST criteria (column 1) or the Choi criteria (column 2).

## CAR T-cell treatment fails if CAR T cells move too slowly

We performed local sensitivity analysis by varying the value of model parameters one at a time across at least one order of magnitude and simulating treatment of tumor type II with intratumoral injection of 5e8 effector CAR T cells (Fig 8). For each parameter, we tested a range of values that encompassed successful and unsuccessful treatment. For each simulation, we calculated the minimum tumor cell count following treatment (the tumor nadir), the timing of the tumor nadir, and the maximum number of CAR T cells. We find that in this model there is a non-monotonic relationship between tumor nadir and the CAR T-cell diffusion constant, $D_C$ (Fig 8A). If CAR T cells diffuse too slowly, CAR T cells kill the core of the tumor but they cannot reach the growth front in order to contain the tumor, and it escapes treatment. At moderately high diffusion constants, the CAR T cells are able to rapidly spread across the whole tumor but maintain sufficiently high intratumoral concentrations to kill tumor cells and proliferate. However, if the CAR T cells diffuse too rapidly, a substantial quantity leak out of the tumor without engaging and killing tumor cells, leaving the tumor intact. This result may not hold in more sophisticated models that explicitly account

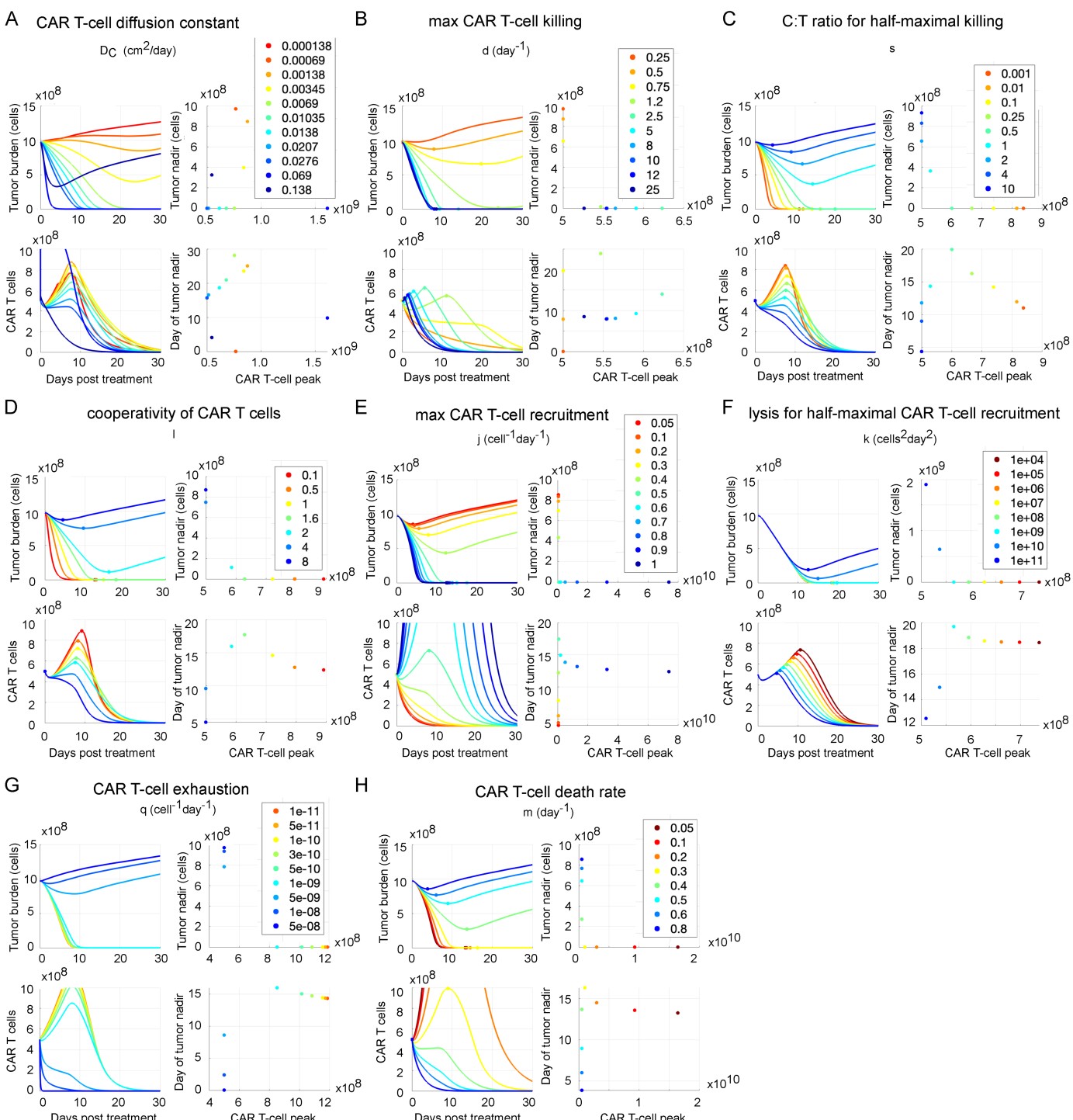

**Fig 8. Tumor and CAR T-cell trajectories for varied values of CAR T-cell parameters.** We varied each model parameter associated with CAR T-cell behavior one at a time across a range of values and simulated intratumoral injection of $5 \times 10^8$ CAR T cells to treat tumor type II out to 30 days post-injection. The resulting trajectories for total tumor cell count and total CAR T-cell count are shown, colored according to the value of the parameter being studied. For each trajectory, we plot the minimum tumor cell count following injection, called the tumor nadir, and the time of tumor nadir as functions of the peak number of CAR T cells. We illustrate the impact of (**A**) the CAR T-cell diffusion constant, $D_C$, (**B**) the maximum CAR T-cell killing rate, $d$, (**C**) the ratio of T cells to tumor cells for half-maximal killing, $s$, (**D**) the cooperativity of CAR T-cell killing, $l$, (**E**) the maximum CAR T-cell proliferation/recruitment rate, $j$, (**F**) the lysis rate for half-maximal CAR T-cell expansion, $k$, (**G**) the CAR T-cell exhaustion rate, $q$, and (D) the CAR T-cell death rate, $m$.

for CAR T-cell binding to tumor cells. Tumor nadir is a monotonic function of the other CAR T parameter values, in agreement with sensitivity analysis of the ODE model studied by Owens and Bozic [45] (Fig 8B–8F). At low values of the maximal CAR T-cell killing rate, d, CAR T cells decline without proliferating due to the structure of the model in which proliferation depends on feedback from tumor cell lysis. Initially, as the CAR T-cell killing rate $d$ increases, the peak CAR T-cell count is higher and occurs earlier. Further increases in CAR T-cell killing capacity still cause the peak to occur earlier, but actually lower the peak because the tumor is eliminated before high numbers of CAR T cells are generated (Fig D Panel B in S1 Text). The CAR T-cell peak was highly sensitive to the maximal CAR T-cell recruitment rate, $j$ (Fig 8E).

## Discussion

In this work we developed a basic reaction-diffusion model for CAR T-cell treatment of solid tumors and characterized the general behavior of the model in a biologically relevant parameter regime. The underlying mechanism that we use for tumor growth is flexible enough to model a wide range of tumor types. This is particularly valuable as there is significant interest in using CAR technology to treat diverse cancers. We further demonstrated that this model captures behaviors observed in preclinical trial data by comparing model predictions with data from mouse-imaging studies tracking tumor and CAR T-cell quantities following local delivery of CAR T cells. We found that accounting for exhausted CAR T cells in model simulations improved the fit to CAR T-cell data from a study that tracked CAR T-cell kinetics. The possibility of exhausted cells in the CAR infusion product also affects the minimum CAR T-cell dose necessary to achieve a given outcome, bringing our results more in line with the reality that current CAR T-cell treatment of solid tumors have been largely unsuccessful [3].

We used our reaction-diffusion framework to compare the response of different tumor types to localized CAR T-cell treatment. In particular we considered dense solid tumors with high proliferation but low diffusivity, moderately aggressive solid tumors with both moderate proliferation and moderate diffusivity, highly-diffuse tumors marked by low proliferation but high diffusivity, and aggressive diffusive tumors with both high proliferation and high diffusivity. Surprisingly, the lowest minimum dose was necessary for the low-proliferation, high diffusivity tumors. That is, assuming the detectable tumor diameter is equal, CAR T-cell therapy was the most effective in treating the tumor with the smallest average density, not the smallest total tumor burden, or the longest VDT. These findings affirm that CAR T cells will not perform equally across different types of solid tumors. It may be particularly promising to pursue CAR T-cell therapy for diffusive tumors. Our analysis suggests that tumor density may be measured when considering the feasibility of CAR T-cell therapy for a particular patient and/or when planning dosages.

There is still a paucity of available data documenting CAR T-cell kinetics in human patients following locoregional delivery to treat solid tumors. However, the dynamics of the total number of CAR T cells predicted by our model reflect qualitative features of CAR T-cell kinetics observed in murine studies of local delivery, namely an early expansion phase followed by contraction [75,88]. In our simulations, almost all successful treatments initiated with a reasonable number of CAR T cells display expansion and contraction phases. Only large doses consisting of fully functional cells administered against small tumors are successful without CAR T-cell count expanding. Liu et al. reviewed the CAR T-cell clinical trial landscape in 2021, which included systemically administered treatments for solid tumors [32]. When comparing responders vs. non-responders across trials, they noted that successful

treatments are marked by a higher peak in CAR T cells and a relatively slower contraction rate compared to unsuccessful treatments [32]. Our model simulations align with this observation from clinical data. As more data becomes available, parameter values for this model should be estimated more rigorously and further extensions to the the model can be incorporated and tested.

We modeled two different modes of local delivery: intratumoral injection vs. intracavitary injection. These options are not always accessible to a patient depending on tumor location and patient health [27]. However, in situations where both are possible, our simulations suggest that intratumoral administration may be successful at lower doses than intracavitary injection. Our results also suggest that for larger tumors, a single dose of CAR T cells administered intracavitarily may induce a partial response or stable disease, but repeat doses of CAR T cells should be explored to move towards tumor eradication. A natural extension of the model would be to optimize the timing of multi-dose regimens.

Using a spatial model is important when studying treatment of solid tumors because assumptions of well-mixed cell populations inherent to previous ODE models do not hold. Furthermore, the movement of CAR T cells within solid tumors has been identified as an important factor in successful treatment. Indeed, Prybutok et al. simulated CAR T-cell therapy using an agent based model and demonstrated that the spatial distribution of tumor cells and healthy cells at the onset of treatment impacted the performance of CAR T cells [91]. Our spatio-temporal model exhibits a failure mode in which there is an initial tumor response to treatment followed by relapse. This clinically observed mode of CAR T-cell failure was not captured by the ODE model from Owens and Bozic [45]. Additionally, as discussed in the above paragraphs, we hypothesize that there may be a potential relationship between tumor density and minimum successful CAR T-cell dose that cannot be captured by an ODE model tracking the total cell populations over time.

This preliminary spatial model has several limitations that could be addressed in future work. The early, Fisher-esque reaction-diffusion models were unable to capture the complex morphology of gliomas observed in experimental and clinical data, motivating incorporation of further biological details. In future work, CAR T cells may be integrated into the numerous more advanced models that exist for gliomas. A natural next step with this model framework could be to include the necrotic core observed in larger tumors and compare the outcome of intratumoral injection at different depths. Extensions of this model could also incorporate a dynamic of the tumor microenvironment, and study the affect of factors like pH, mechanical stress, hypoxia, and IFP on T-cell behavior. We also assume symmetric spherical growth, but angiogenesis (the development of new blood vessels) is needed to support the growth of a tumor beyond the size of about a million cells, at which point the tumor will no longer develop symmetrically. Future work that relaxes the assumption of spherical symmetry and incorporates blood vessels into the model could compare the efficacy of intravascular delivery of CAR T cells, which is also being tested in the clinic notably for hepatic cancers. Relaxing the symmetry assumption will also allow future work to incorporate tumor heterogeneity in terms of target antigen expression into the model, another important mode of CAR T-cell failure. Because of the lack of conclusive data, we assumed CAR T-cell movement is a purely diffusive process. As more data becomes available tracking the distribution of CAR T cells around and within tumors, incorporating chemotaxis or haptotaxis will likely improve the model. This model considers CAR T cells in effector or exhausted states. However, additional T-cell phenotypes such as regulatory T cells could be explicitly modeled to explore their role in treatment outcomes. This level of granularity would be particularly valuable once there is more data tracking T-cell subsets available for model comparison. Partitioning the CAR T cells into short-lived effector cells and long-lived memory cells may also improve the ability

to capture long-term persistence. In clinical data, CAR T cells tend to persist at about $\sim 10\%$ of their peak level [32,92], but in our model they drop to non-detectable levels. However, our model does reflect the fact that CAR T-cell persistence is important to successful treatment. Increasing the CAR T-cell death or inactivation rate causes treatment to lose effectiveness, and conversely lowering these parameters causes treatment to be more effective. Furthermore, future modeling work should incorporate CAR T cells into models of tumor interaction with the endogenous immune system. Though this will substantially increase the model complexity, it will improve predictions over longer time windows, when adaptive immunity is thought to play a key role in the antitumor immune response.

CAR T-cell therapy for solid tumors faces numerous challenges, but significant progress is being made. This work lays out an accessible mathematical modeling framework for studying one possible advance: locoregional delivery of CAR T-cells. Simulation results suggest several testable hypothesis and the model is ready for extension and refinement as more data on CAR T-cell dynamics becomes available.

## Materials and methods

### Numerical analysis

To integrate the PDE given by Eq. (6) we employ Crank-Nicolson [93] for both the tumor and CAR T-cell densities. The use of Crank-Nicolson finite difference methods on an equation with a step-function diffusivity is made possible due to the radial symmetry of the domain. For more complex geometries it will be necessary to employ finite element methods, which we leave for a future study. The standard Crank-Nicolson scheme for the homogeneous diffusion equation,

$$\frac{\partial u}{\partial t} = \nabla \cdot (\nabla u) \tag{8}$$

has an order of convergence of $o\left(\Delta t^2, \Delta r^2\right)$; i.e., second order in time and space. However, the PDE in the present study, (6), would not be expected to maintain the same order of accuracy due to the sharp boundary between proliferative and diffusive regions. Using the method of manufactured solutions we observe a local order of accuracy at $r = 0.5$ of $o\left(\Delta t^{1.5}, \Delta r^{1.1}\right)$ and $o\left(\Delta t^1, \Delta r^{0.6}\right)$ for $u$ and $v$ respectively. Towards the boundary of the computational domain the error increases and we observe a maximum order of accuracy of $o\left(\Delta t^{0.9}, \Delta r^1\right)$ and $o\left(\Delta t^{0.3}, \Delta r^{0.5}\right)$ for $u$ and $v$ respectively. Since the algorithm performs much better towards the center of the computational domain, we choose a domain large enough to keep the extent of the tumor well within the boundary of the domain. The CAR T cells, unlike tumor cells, are free to spread out, but in the absence of tumor cells they no longer proliferate. Consequently, the density of CAR T cells far from the tumor is low in model simulations, as observed in murine experiments. Therefore, although we may detect CAR T cells towards the boundary of the computational domain, their numbers are several orders of magnitude less than around the tumor, and even the CAR T cells near the computational boundary experience an error less than 10% compared to the ground truth according to the manufactured solution. For further details on the numerical analysis, see the supplementary material, S1 Text, page 3.

### Numerical simulations

All results were generated from simulations implemented with MATLAB R2021b. We used a spatial grid with step size $\Delta r = 0.015$ cm and time step $\Delta t = \min(a\Delta r/D_C, \Delta r/5)$ days. This

condition for the time step was empirically chosen to balance efficiency and numerical stability. The extent of the spatial domain was selected based on the initial tumor size and anticipated tumor growth such that the tumor would not reach within 1 cm of the edge of the domain. This is only possible when using a threshold $u^* > 0$ for diffusion. The diffusion threshold keeps the tumor bounded away from the edge of a sufficiently large computational domain, avoiding leakage or having to impose a Neuman boundary condition. All model parameter values used in simulations are included in supplementary Tables A and B in S1 Text.

### Process for fitting model to murine data

We digitized data from two studies on locoregional delivery of CAR T cells using WebPlotDigitizer [94]. The first set of data comes from Fig 4C in Zhao et al. [87], which reports tumor burden prior to and after treatment with a single intratumoral dose of CAR T cells. We fixed the carrying capacity and diffusion threshold for tumor cells, then estimated the tumor growth rate and diffusion constant to minimize the least squares error when comparing the total tumor burden predicted by the model to the pre-treatment data points. We fixed the CAR T-cell diffusion constant at $D_C = 0.0138$ cm$^2$/day as estimated based on Mullazanni et al. [75]. We fixed all remaining CAR T-cell parameters except the maximal lysis rate, $d$, at the values derived from Patient 3 in Owens et al. [45]. We started with an initial guess of $d = 2.25$, from Patient 3 in Owens et al., and increased the value of $d$ until the tumor burden was reduced sufficiently quickly to match the data.

The second set of data came from Fig 4F in Skovgard et al., which reports tumor and CAR T-cell levels measured via BLI following treatment with a single intrapleural dose of CAR T cells [88]. We specifically extracted the mean values for each cell population. In this case, because there are no pre-treatment tumor measurements, we fixed the tumor parameters around those used to define tumor type IV. We fixed the CAR T-cell diffusion constant, the lysis rate for half maximal CAR T-cell proliferation, the CAR T-cell exhaustion rate, and the CAR T-cell death rate at the same values used for the simulation of Zhao et al. data [87]. For the remaining four CAR T-cell parameters, we performed an iterative search to identify a set of values that fit both the tumor and the CAR T-cell data well. We started with initial parameter ranges reported in Table B in S1 Text, then ran simulations with values randomly selected from the parameter space using latin hypercube sampling. We re-centered and narrowed the parameter ranges around the average value from the best 3 runs, and repeated this process until a satisfactory fit to the data was achieved.

### Supporting information

**S1 Text. Supporting information**. S1 Text contains further details on the mathematical model and numerical simulations used to generate the results in this manuscript, as well as the following supporting figures and tables of parameter values.

- **Fig A.** Model-predicted tumor volume doubling time (VDT) and tumor burden at detection as a function of tumor growth parameters.
- **Fig B.** Exhausted cells improve model fit to CAR T-cell data from mouse imaging study.
- **Fig C.** Outcome maps given varied tumor size at the time of treatment.
- **Fig D.** Outcome maps given varied time of patient evaluation
- **Fig E.** Relating quantitative tumor characteristics to minimum CAR T-cell dose for tumor eradication
- **Fig F.** Convergence analysis via method of manufactured solutions

- **Table A.** Parameter values used in numerical simulations of the 4 tumor types described in Section 2.2
- **Table B.** Estimated model parameters to fit model output to murine study data

(PDF)

**S1 Data. Dataset 1.** Data digitized from pre-clinical mouse studies and used for model fitting. (.XLSX)

## Author contributions

**Conceptualization:** Katherine Owens, Aminur Rahman, Ivana Bozic.

**Data curation:** Katherine Owens.

**Formal analysis:** Katherine Owens, Aminur Rahman.

**Investigation:** Katherine Owens.

**Methodology:** Katherine Owens, Aminur Rahman, Ivana Bozic.

**Project administration:** Katherine Owens.

**Software:** Katherine Owens, Aminur Rahman.

**Supervision:** Ivana Bozic.

**Validation:** Katherine Owens.

**Visualization:** Katherine Owens, Aminur Rahman.

**Writing – original draft:** Katherine Owens, Aminur Rahman.

**Writing – review & editing:** Katherine Owens, Aminur Rahman, Ivana Bozic.

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
