## [Decision Letter · Decision Letter 0]

19 Jan 2025

PCOMPBIOL-D-24-01849

Spatiotemporal dynamics of tumor - CAR T-cell interaction following local administration in solid cancers

PLOS Computational Biology

Dear Dr. Owens,

Thank you for submitting your manuscript to PLOS Computational Biology. After careful consideration, we feel that it has merit but does not fully meet PLOS Computational Biology's publication criteria as it currently stands. Therefore, we invite you to submit a revised version of the manuscript that addresses the points raised during the review process. One of the reviewers suggests moving the Methods section before Results. As this is an option that the journal permits, we leave this choice up to your discretion but please provide a rationale for whatever changes you decide to make.

Please submit your revised manuscript within 30 days Mar 21 2025 11:59PM. If you will need more time than this to complete your revisions, please reply to this message or contact the journal office at ploscompbiol@plos.org. Please include the following items when submitting your revised manuscript:

We look forward to receiving your revised manuscript.

Kind regards,

James R Faeder

Academic Editor

PLOS Computational Biology

Tobias Bollenbach

Section Editor

PLOS Computational Biology

**Journal Requirements:**

3) We have noticed that you have a list of Supporting Information legends (S2 File. Dataset 1) in your manuscript. However, there are no corresponding files uploaded to the submission . Please upload them as separate files with the item type 'Supporting Information'.

4) We noted that subfigures S5 (E-H) are labeled as Figure S6. Please check the labels of the supplementary figures.

Potential Copyright Issues:

i) Figures 1B, 1C. Please confirm whether you drew the images / clip-art within the figure panels by hand. If you did not draw the images, please provide (a) a link to the source of the images or icons and their license / terms of use; or (b) written permission from the copyright holder to publish the images or icons under our CC BY 4.0 license. Alternatively, you may replace the images with open source alternatives. See these open source resources you may use to replace images / clip-art:

**Reviewers' comments:**

Reviewer's Responses to Questions

**Comments to the Authors:**

**Please note that one of the reviews is uploaded as an attachment.**

Reviewer #1: PCOMPBIOL-D-24-01849

Spatiotemporal dynamics of tumor - CAR T-cell interaction following local administration in solid cancers

In this article by Owens, Rahman, and Bozic, the authors present a mathematical model to study two different routes of administration of CAR T cells in solid tumors. They use previously published preclinical data and prior literature to ground their model in data and to study different possible therapeutic outcomes. The manuscript is well written, well motivated, and is likely to be of interest to a wide audience in the fields of mathematical biology, cancer research, and CAR T cells.

The authors principal conclusion is that “We demonstrate that the model can recapitulate tumor and CAR T-cell data from small imaging studies of local administration of CAR T cells in mouse models.”, and that “locally administered CAR T cells will be most successful against slowly proliferating, highly diffusive tumors.” I have concerns about both of these conclusions, namely 1) their model is fit to imaging data, and therefore does not recapitulate the dynamics per say as much as reproduce them, and 2) the efficacy against highly diffusive tumors may be an artifact of the diffusion thresholding in their model. However, there are plenty of insights from their model that can be well justified and supported as conclusions.

I have a few significant concerns regarding the modeling:

Why do the authors threshold diffusion? This is a major assumption in the model that is not consistent with previous PDE modeling in this area, and is explained or justified at all.

The authors glaze over the exhausted cell compartment, and provide no rationale for its inclusion or the key parameter (q) in determining the fraction. Also, exhausted cells are not graphed or discussed in detail. So why include them?

Similarly, the D function (after equation 3b) is complex and not motivated at all. F2 appears to be a very interesting and complex term. Please explain to the reader.

I think the assumption of uniform CAR T cell density on the tumor surface (even in spherical symmetry) is far too simplified, but I do not insist it be changed. Consider providing much more justification for this.

Why do the authors present 4 tumor growth archetypes but present only 3 in figures 3,4? Confusing.

I think the manuscript would benefit greatly from a discussion about the parameters and their meanings, particularly the CAR parameters. Can the tumor growth/diffusion and CAR dynamics reveal more specific/quantitative relationships between the parameters? Also, the 20% reduction in parameters to model ‘reduced efficacy in solid tumors’ is important and not studied or motivated.

Minor comments:

Check carefully for typos and latex issues. There are more than a few. (ex. T-cells vs. T cells).

Numerical values in caption of Figure 2 are cumbersome to read. They would be better as a table.

I find figure 1A to be more confusing than helpful. Difficult to interpret the 4 arrows and dots, and the complex term at the top (jD(u,v)^2v/, …) is not explained in the caption or main text in sufficient detail to understand.

Can the variance across mice be shown in Figure 5? Authors state the model was fit to the mean value.

In vitro and ex vivo used to mean the same thing. Consider using a single term.

The \cdot notation used in place of \nabla is unconventional and also not introduced or explained (equations 1,2).

Reviewer #2: Uploaded as an attachment

**Have the authors made all data and (if applicable) computational code underlying the findings in their manuscript fully available?**

Reviewer #1: Yes

Reviewer #2: Yes

PLOS authors have the option to publish the peer review history of their article (what does this mean?). If published, this will include your full peer review and any attached files.

Reviewer #1: No

Reviewer #2: No

**Figure resubmission:**
---

## [Decision Letter · Decision Letter 1]

6 May 2025

Dear Dr. Bozic,

We are pleased to inform you that your manuscript 'Spatiotemporal dynamics of tumor - CAR T-cell interaction following local administration in solid cancers' has been provisionally accepted for publication in PLOS Computational Biology.

Best regards,

James R Faeder

Section Editor

PLOS Computational Biology

Tobias Bollenbach

Section Editor

PLOS Computational Biology

Reviewer's Responses to Questions

**Comments to the Authors:**

Reviewer #1: Thank you for thoroughly addressing all reviewer concerns. My only remaining comment is that the model schematic in figure 1 would be helpful to include, provided that it is consistently annotated (or not). In the original version, not all interactions were labeled, and the labels that were provided were not described. I think a schematic of the model, even a simplified one, would benefit the paper and reader.

**Have the authors made all data and (if applicable) computational code underlying the findings in their manuscript fully available?**

Reviewer #1: Yes

PLOS authors have the option to publish the peer review history of their article (what does this mean?). If published, this will include your full peer review and any attached files.

Reviewer #1: No